# Slope Dynamics in Relation to the Occupation and Abandonment of a Mountain Farm in Þistilfjörður, Northeast Iceland

Julien Lebrun [1], Najat Bhiry [1,*] , James Woollett [2] and Þorsteinn Sæmundsson [3]

1   Département de Géographie and Centre D'Études Nordiques, Université Laval,
    Québec, QC G1V 0A6, Canada
2   Département des Sciences Historiques and Centre D'Études Nordiques, Université Laval,
    Québec, QC G1V 0A6, Canada
3   Institute of Earth Sciences, Faculty of Life- and Environmental, Sciences, University of Iceland, Sturlugata 7,
    101 Reykjavík, Iceland
*   Correspondence: najat.bhiry@cen.ulaval.ca

**Abstract:** Extreme weather events such as storms, heavy snow accumulation, rapid snowmelt, and heavy rain have been closely related to slope instability in arctic and subarctic regions. In this paper, we investigate the historical activity of slope processes such as snow avalanches and debris flows in Þistilfjörður, northeastern Iceland, and examine their possible role in the occupation and abandonment of three archaeological sites located on slopes of Mt. Flautafell. The study combines geomorphological and stratigraphical surveys with historical records, notably *Jarðabók Árna Magnússonar og Páls Vídalíns* and *Sýslu og sóknarlýsingar Bókmenntafélagsins Svalbarðssókn*. Geomorphological surveys show numerous features that are indicative of active slope movement processes in and around the investigated sites. Our results suggest that the slopes experienced periods of instability during the occupation of these sites. The burial or destruction of some parts of the homefield at the Flautafell farm reveals slope activity, which may also be related, at least indirectly, to the abandonment of the farm at Norður Hús sometime before A.D. 1300. Nearby auxiliary farm installations of Stekkur remained untouched by slope processes even though they are situated in a vulnerable area. Further study and dating of slope processes and farm occupation could allow them to be used as proxies for deteriorating environmental conditions affecting the region.

**Keywords:** slope processes; northeastern Iceland; farm occupation and abandonment; geomorphological and stratigraphical surveys; historical records

## 1. Introduction

Iceland is a geologically and geomorphologically active island whose regional-scale landscape is the result of glacial, volcanic, and tectonic processes [1,2]. At the local scale, aeolian, fluvial, and slope processes combine to produce very dynamic landscapes. In addition, the island's position in the North Atlantic region creates a climate characterized by frequent, abrupt variations and anomalies [3,4]. These climate conditions caused significant environmental changes during the Holocene, which are recorded in the landscape and can be used as proxies to understand the evolution of the environment. For example, glacial fluctuations of the Icelandic Ice Sheet (IIS) are consistent with significant changes in regional climate throughout the Holocene [5].

Slope processes are closely associated with climate fluctuations, and specifically with extreme meteorological events, in the Arctic and subarctic regions. In Iceland, extreme meteorological conditions (e.g., significant rainfall, rapid snowmelt, heavy snow accumulation, and winter storms) have been identified as triggering factors for debris flows and snow avalanches [6–8]. The resulting colluvial deposits can be considered as proxy

records of extreme meteorological events. Snow avalanche and debris flow events have major local-scale impacts on landscapes and can pose major risks for human settlement and activity in many other areas of the world with similar climate conditions, such as Norway and the Faroe Islands (e.g., [9–11]).

Since the *Landnám* period of colonization, beginning around A.D. 877 $\pm$ 1 [12], numerous documented episodes of slope processes have caused considerable damage to human infrastructures and have caused many casualties [13,14]. One such example is mentioned in the saga *Sturlunga*, written in A.D. 1118, the first known text in Iceland to report a snow avalanche episode [15]. Since then, until the present day, approximately 680 casualties derived from slope process events have been documented, a remarkable number for such a sparsely populated country [14]. While these anecdotal sources tend to provide evidence of only major events near human settlements, the widespread distribution of geomorphological events and the climatic circumstances suggest that slope processes are a major and constant factor in Iceland's landscape development at the regional scale and in general.

Although many geomorphological studies in Iceland have investigated slope dynamics from the perspective of natural hazard prevention and mitigation [16], little research has investigated the impact of slope dynamics on past farm occupation and abandonment. Studies of slope dynamics have been concentrated in the mountainous fjords of western, northern, and eastern Iceland, where gravity processes present the greatest risk for human infrastructure and local populations (e.g., [16,17]). Historical records of mass movement episodes in Iceland were compiled by Jónsson [18] and were later mapped by Björnsson [13]; however, the data suggest that very few movement episodes have been reported in the northeast region. This contradicts the geomorphological evidence that shows conspicuous features of slope activity, for example on the Flautafell mountain, which is the object of the present study [19].

The Norse colonization of northeastern Iceland at about A.D. 1000 took place in a very dynamic natural setting. The landscape of settlement and its evolution have been widely discussed by archaeologists, geoarcheologists, and paleoecologists (e.g., [19–24]) who have noted the importance of several anthropogenic landscape impacts including widespread cutting of wood and deforestation, manipulation of drainage, and accelerated soil erosion. However, the impact of slope processes on the landscape and on human occupation is not well understood. Although fieldwork and historical evidence from the 18th century indicate the existence of fields that were destroyed by debris-flow activity [25,26], it is not clear how this affected land management. This study is part of a large multidisciplinary program that aims to increase our archeological, paleoeconomic and palaeoenvironmental knowledge in the Svalbarðstunga study region (valleys of Svalbarðsá and Sandá rivers) at the local and regional scales (e.g., [24,27–29]). The project examines many factors that could be linked to farm occupation/abandonment and land management, such as: social, economic, and agricultural practices, climate change, natural hazards, epidemics, and disease. The most recent paleoecological results showed that ecological changes in Svalbarðstunga linked to the Norse and medieval settlement were more limited than in the rest of Iceland, where woodland clearance and erosion have been associated with human settlement [24,28,30,31].

The main objective of our study is to document the relationship between slope processes and farm occupation at the foot of the Flautafell mountain (at the western limit of Svalbarðstunga) using a combination of geomorphological analysis and the review of historical documents. This study also aims to document the occupation history of marginal upland farms.

## 2. Materials and Methods

### 2.1. General Setting and Study Site

The study area is part of the Svalbarðsá river valley, located in the western part of Þistilfjörður, Northeast Iceland (Figure 1). It is the coldest region in Iceland, with a mean annual temperature that is 1.8 °C lower than the observed temperature in the capital, Reykjavik. This temperature difference is due to predominant Arctic influences such as

sea ice and polar air masses derived from the marine polar front and the East Greenland Current [32,33]. Previous geomorphological and archeological work concentrated on the bottom of the valley (e.g., [23,27–29]), while the present study examines slope processes affecting settlement on the mountain slopes comprising the western boundary of the valley. We primarily focused on the slopes of Mt. Flautafell (522 m a.s.l.), which is located about 7 km inland from the coast. At the foot of the mountain, a series of river terraces separate the slopes and the colluvial system from the alluvial plain of the Svalbarðsá river. The slopes are marked by numerous geomorphological features resulting from slope processes such as avalanche boulder tongues and colluvial cones. In the apical part of the slope, an approximately 30 m high basaltic rock wall separates the slope from the plateau. The rock wall is incised by numerous gullies and channels in which water and debris are stored and concentrated.

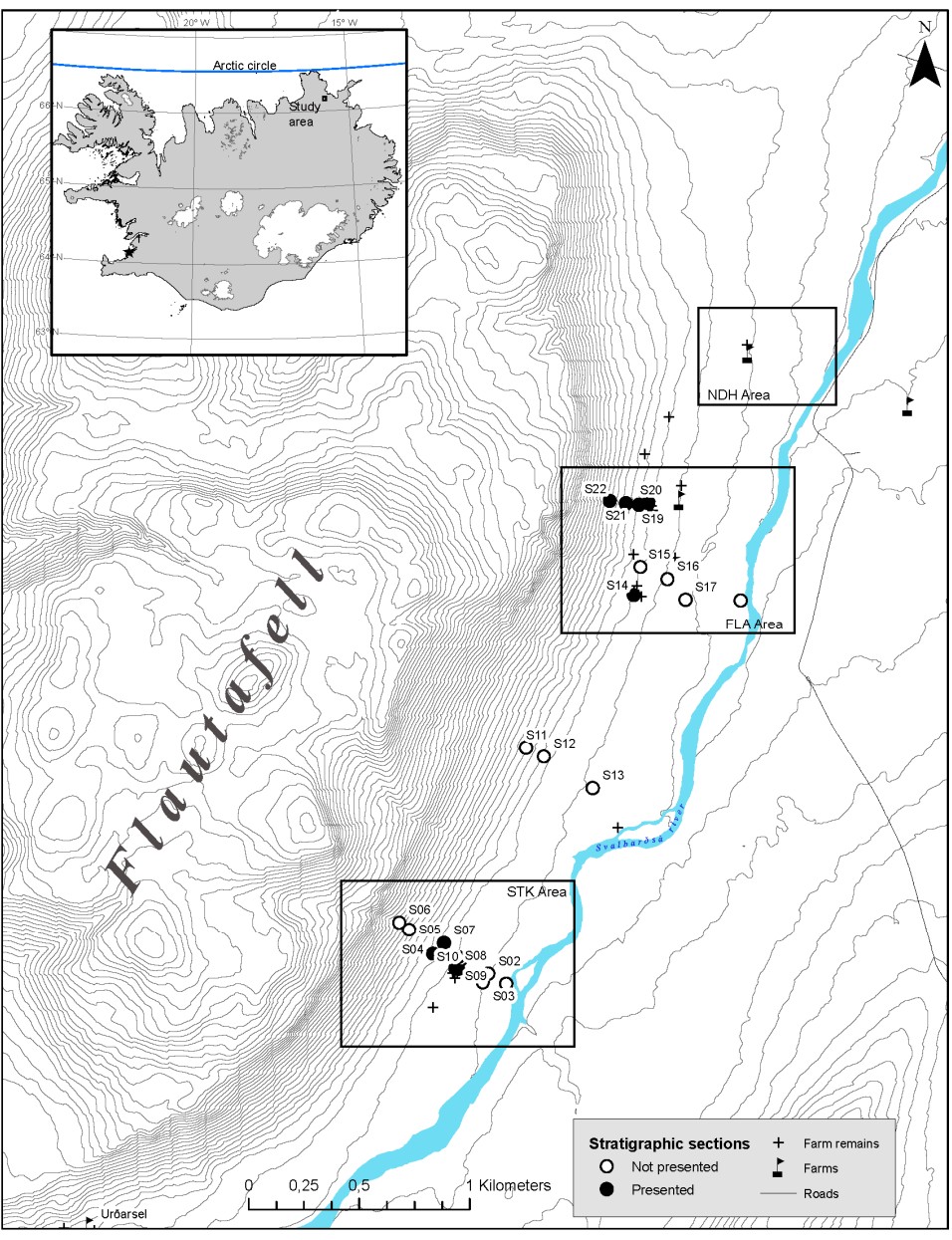

**Figure 1.** Location of the study area in northeastern Iceland and the location of sites used for stratigraphical study on the slopes of Mt. Flautafell. The three sectors in the study, Flautafell farm (FLA), Stekkur (STK), and Norður Hús (NDH), are delineated by the squares.

### 2.1.1. Physical Characteristics of the Area

The climate in the study region is classified as polar tundra (ET) according to the Köppen classification [32]. Mean monthly temperatures recorded in Raufarhöfn range from −1.6 °C in January to 8.4 °C in August [34]. Annual precipitation averages 732 mm, with 61% falling as snow.

Local vegetation is formed of grass and sedge with diverse shrubs (dwarf birch, Arctic willow, dwarf willow), mosses, and lichen. Although vegetation occupies most of the landscape, erosion patches are observed on the most exposed areas.

The local landscape consists of two main geological units: the low-lying areas are formed of tertiary basalt and andesite (3.1 Ma), while the mountains consist of hyaloclastite and tuffaceous sediment formed by subglacial volcanic activity during the Pleistocene (0.7 Ma) [2,35]. Glacial till forms a hummocky moraine, and alluvial sediments constitute the bulk of the surficial deposits in the research area. The topography consists of low rolling hills up to 10 m high with bare summits caused by aeolian erosion and depressions occupied by peatlands.

### 2.1.2. Study Sites

This study focuses on three main farm sites: (a) the Flautafell farm (FLA); (b) the abandoned Norður Hús farm site (NDH); and (c) a group of unnamed archaeological ruins consisting of a sheepfold and hay-drying platforms, termed Stekkur (STK) for the purposes of this study (Figures 1 and 2).

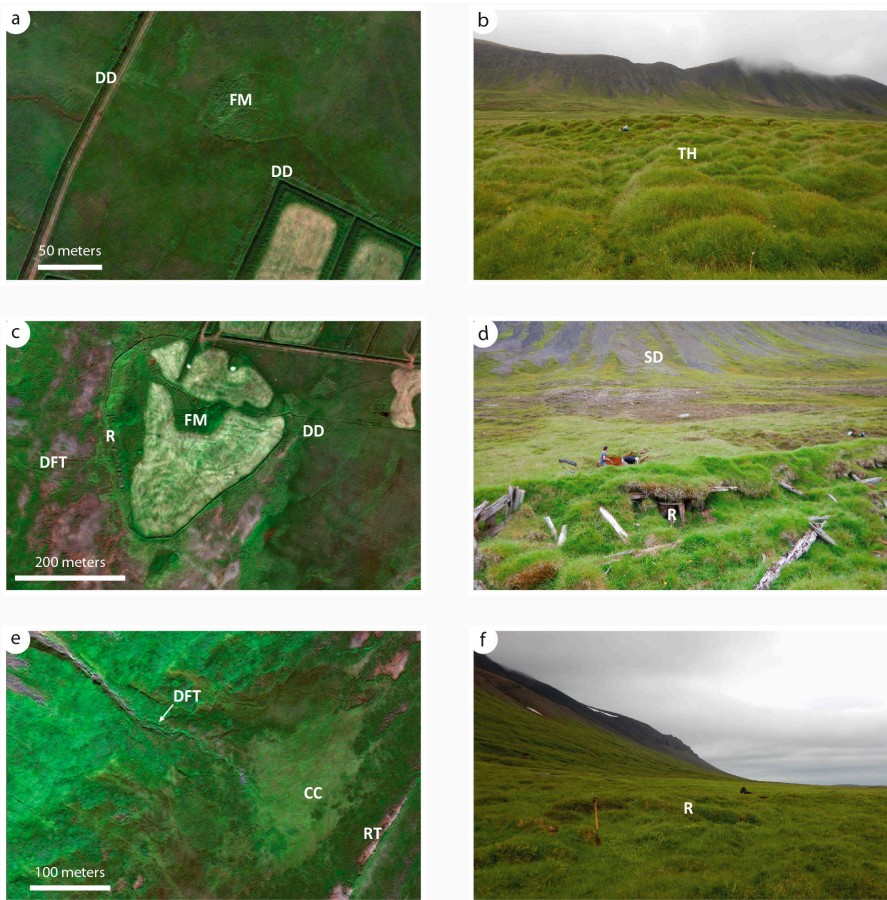

**Figure 2. Left**: satellite images of the three study areas; **Right**: view from the farms relative to the slopes of Mt. Flautafell from the ground. (**a**,**b**): Norður Hús, (**c**,**d**): Flautafell, (**e**,**f**): Stekkur. Source of satellite imagery: DigitalGlobe. DD: drainage ditches, FM: farm mound, TH: Thúfur, DFT: debris-flow track, R: ruins, SD: slope deposits, CC: colluvial cone, RT: river terrace.

The principal study site (80 m a.s.l.) is the location of the historical farmstead of Flautafell, which was occupied sometime prior to A.D. 1300 until 1978 [36]. The site extends close to the foot of the slope and encompasses meadows and small peatlands (Figure 3). The farm site consists of a tell-like farm mound comprising ruined turf buildings used between circa 1300 and the mid-20th century, the ruins of turf out-buildings, and a hay field bounded by a stone and turf wall. Although it is somewhat humid, the hay field is better drained than the surrounding pasture lands and is relatively productive. The neighboring farm at Garður still uses the field for hay harvesting and sheep grazing. A principal ditch that channels water downslope through the field beside the farm mound, and several secondary ditches dispersed throughout the field provide evidence of efforts to drain the hayfield. Along the western (upslope) margin of the field, there is a row of stone-walled buildings located less than 20 m from the inflexion point of the talus slope. There is a clear overlap between the slope deposits, the field, and the farm buildings in the area. Several smaller abandoned turf structures lie on the outskirts of the hayfield, and at the southern tip of the field, a flat fluvial sandy terrace marks the place where the most recent farm house stood in the mid-20th century.

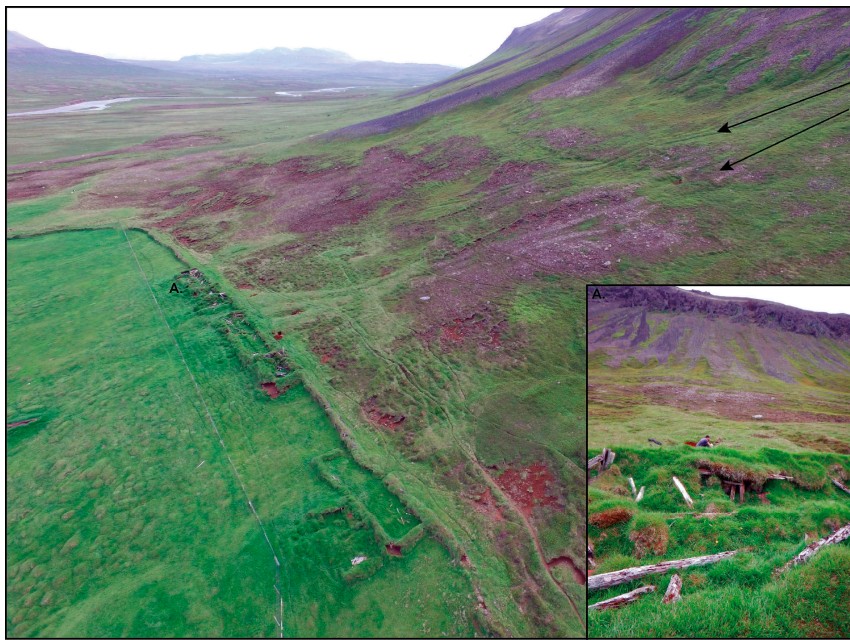

**Figure 3.** Aerial view from a drone looking towards the south of the western part of the field at the Flautafell farm. (A) A view from the ruins looking towards the slope. Farm ruins are visible in the foreground and snow avalanche landforms can be seen in the background.

Norður Hús (64 m a.s.l.) consists of a substantial farm mound with ruins of turf buildings, the ruins of a second adjacent turf building, and at least one additional, small, isolated turf ruin, possibly an outbuilding, on a low mound. This group of ruins is located 0.5 km to the northeast of the Flautafell homefield, in a very boggy hayfield that presents major cryoturbation features such as thúfur (hummocks) (Figure 2a,b). Since it is common practice for farmers to eliminate unused structures from their hayfields, it is possible that this finding suggests that the field has not been used for hay production for many years. The ruins themselves are indistinct, implying considerable age. The area is located 650 m downstream (south) from a large corridor in the mountainside with considerable debris-flow activity connected to a catchment on the plateau. Multiple streams flow from this catchment, one of which is directly linked to the Norður Hús farmstead. A narrow, irregular, and largely infilled ditch bounds the northern, eastern, and western sides of the farm mound, demonstrating efforts to drain the field, probably during the site's occupation. Several additional, deep, modern drainage ditches have also been dug with mechanical

equipment in and around the wet hayfield. One of these ditches is partially filled in, indicating ongoing recent sedimentary transport from higher ground.

Stekkur is the most inland site in this study and the highest site, at 120 m a.s.l. It is located 2.25 km south of the Flautafell farm site (Figure 1), halfway to the neighboring farm of Urðarsel, which was itself abandoned during the early 20th century (see Þórarinsson et al. [36]). The site has no known local toponym and was named Stekkur for the purposes of this study, since "stekkur" is an Icelandic term for a small enclosure used as an animal pen. The use of such sites was typically seasonal and related to pastoral use such as milking ewes. The site comprises four small and low turf and stone structures, each of which lacks a roof and a clear entrance, and that resemble traditional sheep milking or hay storage enclosures (Figure 2e,f). The site's location on a grassy, steep, and relatively well-drained colluvial cone on the mountain side suggests that the location was chosen for its suitability for harvesting and storing winter fodder rather than habitation. An incised channel originates below the rock wall and channels the intermittent flow northwards on the side opposite to the structures (Figure 2e).

*2.2. Methods*

The combination of geomorphological analysis with the study of historical documents provides an original perspective on the history of the area. Limited archaeological interventions were used when examining the ruins and dating human occupation. Techniques included soil core prospection and documentation of stratigraphy in existing erosion channels and drainage trenches.

2.2.1. Geomorphological and Stratigraphical Analyses

Geomorphological surveys of the southeastern slope of Mt. Flautafell at the three archeological sites were carried out in the summers of 2016 and 2017. The distinction we made between landslide, debris flow, and snow avalanche deposits and for boulder distribution diagnostic was based on Blikra and Nemec [10] and Decaulne et al. [16]. Field measurements were taken along multiple longitudinal and lateral profiles using a differential global positioning system (DGPS). Profiles corresponding to typical landforms found on the slopes of Mt. Flautafell were selected in order to measure their topography. Each point was manually recorded at a mean interval of 2 m with a resolution of 3 cm. Data were processed and analyzed using Excel. Stratigraphical sections were excavated on the distal, medial, and proximal parts of the slope in order to document the history of slope sedimentation. In addition, sections were excavated outside the homefield about 5–6 m from ruins to assess the impact of slope processes on human infrastructures. Each section was subdivided into units based on the grain size of the deposits, their composition (mineral, organic matter, or tephra), color, and sedimentary structures.

Five tephra layers were identified in the study area [27]. These include (a) two thick, light-colored silicic layers, which are identified as Hekla 4 (H4; 3825 ± 15 cal. y BP) and Hekla 3 (H3; 2880 ± 35 cal. y BP), and (b) three thin, dark, basaltic layers, which are identified as the Landnám tephra layer (LTL; A.D. 877 ± 1) (which is one of the three tephras found at 58% of early settlement sites), the V-Sv tephra (A.D. 938 ± 6), and the Eldgjá tephra (A.D. 938) [10]. The other tephras found in the study area include Hekla 1300 (H1300; A.D. 1300) and Veiðivötn 1477 (V1477; A.D. 1477) [12,37]. Several tephras were found in excavated cross-sections during this study and were sampled and analyzed at the NORDVULK laboratory at the University of Iceland. The results are reported this paper in A.D./B.C.

An Oakfield Soil corer with a $\frac{3}{4}$ inch tube was used to conduct soil core tests in and around farm mounds, building ruins, and hayfields. Soil core tests were used to detect traces of human occupation and to evaluate their stratigraphy. Relative chronologies of occupation were established according to onsite visual identification of the main tephras observed in the research area and through identification of sediment layers, incorporating anthropogenic inclusions such as charcoal, peat ash, and animal bones [38,39].

2.2.2. Archives

Historical documents were used to provide insights into the occurrence of slope processes, their location, and origin, and to aid in the interpretation of their impact on human occupation. We identified useful source documents through study of historical records concerning landslides, snow avalanches, and similar events in Iceland compiled by Jónsson [18]. Two documents were particularly useful for the present study: *Jarðabók Árna Magnússonar og Páls Vídalíns* and *Sýslu og sóknarlýsingar Bókmenntafélagsins Svalbarðssókn*. The first document was a tax census compiled by Árni Magnússon and Páll Vídalín, who were commissioned by the Danish colonial administration to travel the country and create a complete land and stock register for all properties. This first document of Iceland's conditions provides a very rich and comprehensive record of Iceland's social, economic, and political circumstances (see McGovern [22]).

The second document was produced in the 1840s as part of a socioeconomic survey of Iceland carried out by Jónas Hallgrímsson and the Icelandic Society of Letters [40]. Its main objective was to create a social and economic portrait of Iceland and its districts based on responses to a survey that had been sent out to parish priests across the country, including that of Svalbarðssókn, Flautafell's parish. Additional data regarding the Flautafell farm in the 18th to 20th centuries was obtained from the *Manntall* and *Land og Folk* farm registers. Historical maps compiled by the Danish Geodetisk Institute in 1821 and afterwards were used to verify the locations of farmsteads.

## 3. Results

### 3.1. Geomorphological Survey of the Flautafell Area (FLA)

Sedimentary loads of large, highly angular clasts are typically associated with snow avalanches or debris flows. Several avalanche road bank tongues are clearly visible upslope from the *Flautafell* farm located at the foot of the slope (Figure 4). These landforms may indicate past recurrent snow avalanche activity at the site. The recent and present-day scree or debris flow landforms are located a few hundred meters upslope from the farm wall (Figure 4). However, many scattered snow avalanche boulders occur at the base of the slope close to the farm. Based on this geomorphological evidence, the farm is clearly located in a vulnerable position that exposes it to the impact of snow avalanches.

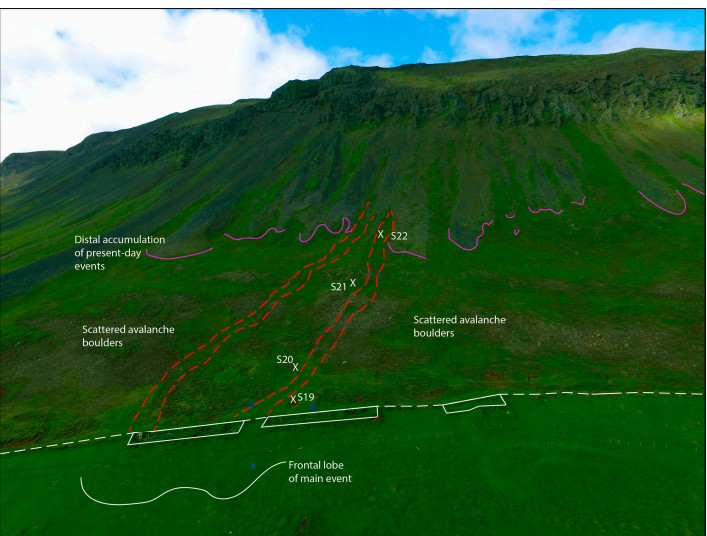

**Figure 4.** Aerial view of the western part of the Flautafell farmland and the slope above. The dashed red line shows the trajectory of the two channels originating between avalanche boulder tongues and heading towards the farm. The extent of active avalanche boulder tongues is delineated in pink, farm outbuildings are represented by a solid white line, and the homefield wall is shown by a dashed white line.

Two relic channels were observed along the western wall delineating the field of the farm (Figures 4 and 5). These channels originate from a depression located between two avalanche boulder tongues. The intersection of the channels and the field forms a hummocky topography. Coring in this area exposed angular blocky deposits associated with slope processes. This area was interpreted as the depositional lobe of the two channels. The channel and levee morphology suggest a slope process with a high water content, such as a debris flow or a slush flow. The farm ruins are located on top of this terrain feature and are relatively well preserved. Therefore, the buildings are younger than the underlying mass-movement deposits.

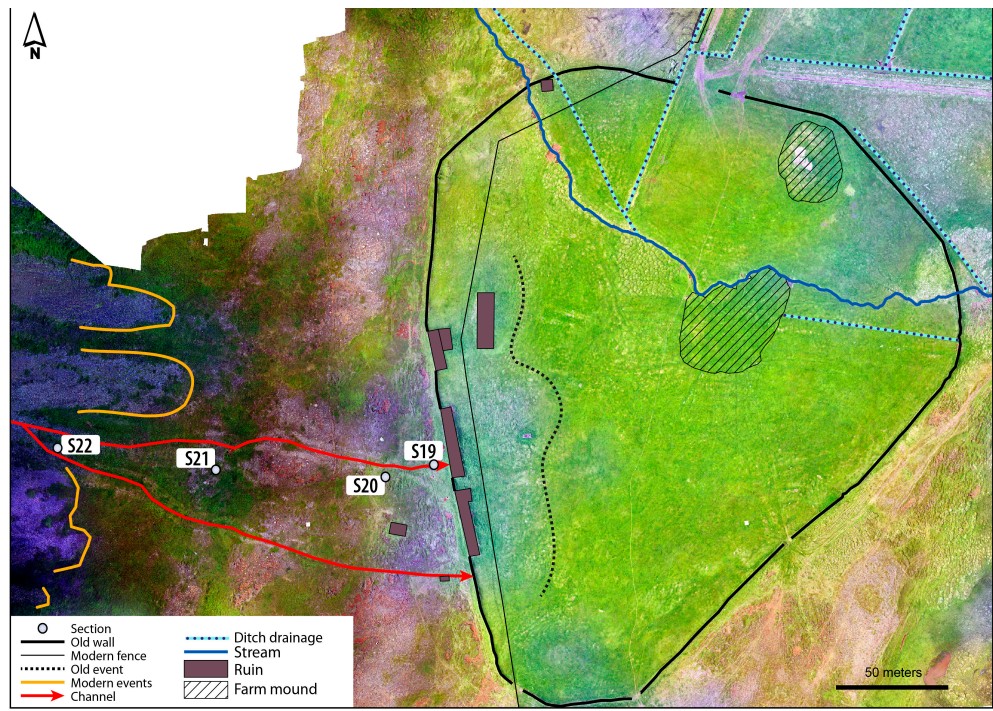

**Figure 5.** Aerial view of the Flautafell farm (FLA) showing the distal part of the avalanche boulder tongues (white lines) and two channels (red lines) reaching the western wall of the farm and its ruins.

*3.2. Analysis of Stratigraphic Sections and Historical Data*

3.2.1. The Flautafell Farm Area (FLA)

Stratigraphical sections were excavated along one of the channels to identify the sedimentary processes involved and their formation. Section S19 is in the northern channel, a few meters outside the western wall and the adjacent ruins (Figure 5). It is formed of alternating coarse blocky deposits and fine silty sand (Figure 6). At the bottom, Facies IIa is clast-supported and shows no stratification or structure. It is composed of subrounded and angular blocks with a matrix of coarse sand and subrounded gravel. The very heterogenous nature of the sediments appears to be the result of two different processes. First, the Svalbarðsá River deposited alluvial sediments that are well-rounded due to fluvial transport. Second, the deposits were then disturbed by a mass movement episode that resulted in the deposition of angular blocks. Facies I is represented by massive reddish brown silty sand that appears to be paleosoil. It includes a yellowish silicic tephra identified as H3 (1050 B.C.). This tephra layer has been disturbed by cryogenic processes, slope wash or wind, or a combination of these processes. It also includes two dark basaltic tephras: H1300 and V1477. Above, a coarse massive deposit composed of angular to subangular basaltic blocks forms Facies IIb. Its structure is clast-supported with a matrix of organic-rich fine sand; it is overlain by a modern soil.

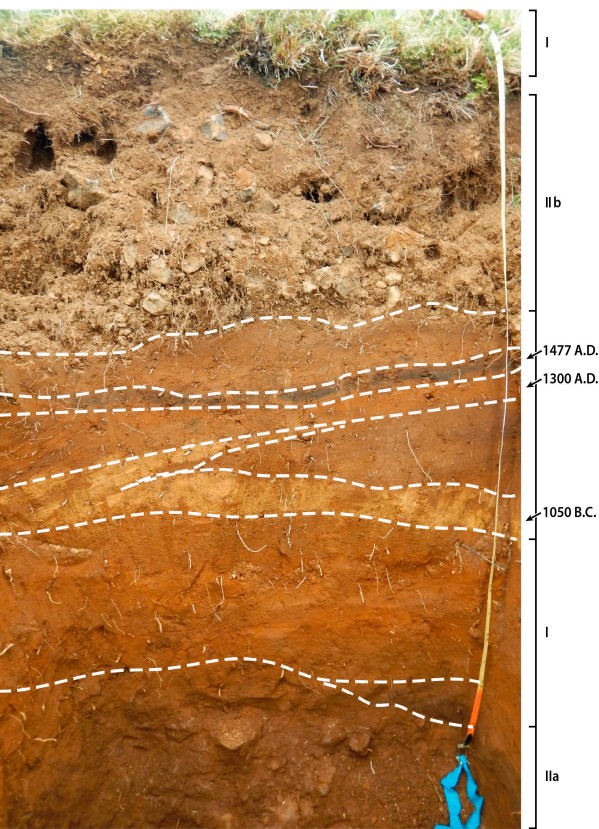

**Figure 6.** Stratigraphical section S19 at the edge of Flautafell's western homefield wall. Slope deposits (Facies IIa and Facies IIb) are separated by fine sediments (Facies I) with dark- and light-colored tephras. The scale on the right (tape measure) is one meter.

In summary, section S19 shows evidence of slope activity prior to ca. 1050 B.C. (3000 B.P.), followed by a period of stability which lasted until at least A.D. 1477. One or more episodes of slope processes occurred after A.D. 1477. This sequence and chronology of slope activity and stability has also been observed in sections excavated upslope from the farm, and slope activity after A.D. 1477 was also clearly identified in other sections on the slopes of Mt. Flautafell (Figure 7).

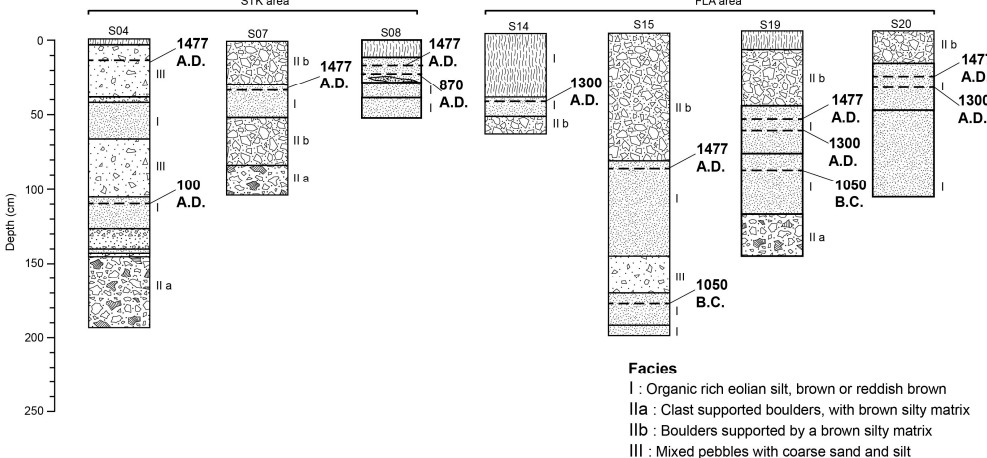

**Figure 7.** Synthesis of stratigraphical profiles showing alternating phases of stability (Facies I) and activity (Facies II, III, IV) in Flautafell, Northeast Iceland. For cross-section localization, see Figure 1.

According to historical evidence compiled by ormóðsson (1971), corroborated by *Byggð í iistilfirði*, the Flautafell farm was occupied as early as A.D. 1318. By A.D. 1712, the

farm had been abandoned for an extended time along with several other farmsteads in the area [41]. However, no reasons were reported for the abandonment of these farms. By A.D. 1712, the farm was reoccupied. Other short periods of abandonment can be inferred from the absence of references to the farm in land records in A.D. 1748 and A.D. 1762. This suggests two short periods of abandonment sometime between A.D. 1735–1754 and A.D. 1760–1785. Finally, the site was abandoned once again in the 1970s, and its land is now used by the neighboring farm of Garður.

The historical documents also indicate numerous mass-movement episodes associated with the Flautafell farm (Table S1). First, the *Jarðabók Árna Magnússonar og Páls Vídalíns* indicates that the Flautafell field was untended after being partially destroyed by a landslide and subsequently abandoned. It is also said that the farm had been named "Flutningsfell". In Icelandic, the term *"Flutning"* carries a meaning of sudden transport or movement, such as the flight of a ground-dwelling bird. For this reason, the farm at Flautafell (and the mountain itself) was known as Flutningsfell or the "moving mountain", which we interpret as a reference to the slope processes occurring in the area. There is no date for this landslide offered in historical sources.

Second, *Sýslu og sóknalýsingar Bókmenntafélagsins Svalbarðssókn* states that the Flautafell farm possesses little grassland because it had mostly been destroyed by landslides. It also adds that there are traces of a "landslide" close to the homefield wall. It should be noted that what is designated as landslide features in the text presents a debris flow morphology in the field. This might be related to the two debris flow channels we observed (Figures 4 and 5); however, it is impossible to confirm the precise location of the elements described in the document because there is insufficient biophysical description of the area.

The homefield stratigraphy as observed in a ditch section is formed by a series of highly decomposed peat and organic matter layers that contain charcoal, wood and peat ash, and calcined animal bone (Figure 8). These elements likely correspond to detritus from human occupation that was deposited as garbage or spread as fertilizer in the homefield. Anthropogenic deposits were first observed underneath the H1300 tephra but are present throughout the sections without any clear subsequent breaks (Figure 8). The uppermost layer (post A.D. 1300) is formed by organic-rich aeolian silt (soil or paleosol). This suggests that the farm was occupied almost continuously from circa A.D. 1300 until the mid-20th century with short periods of abandonment.

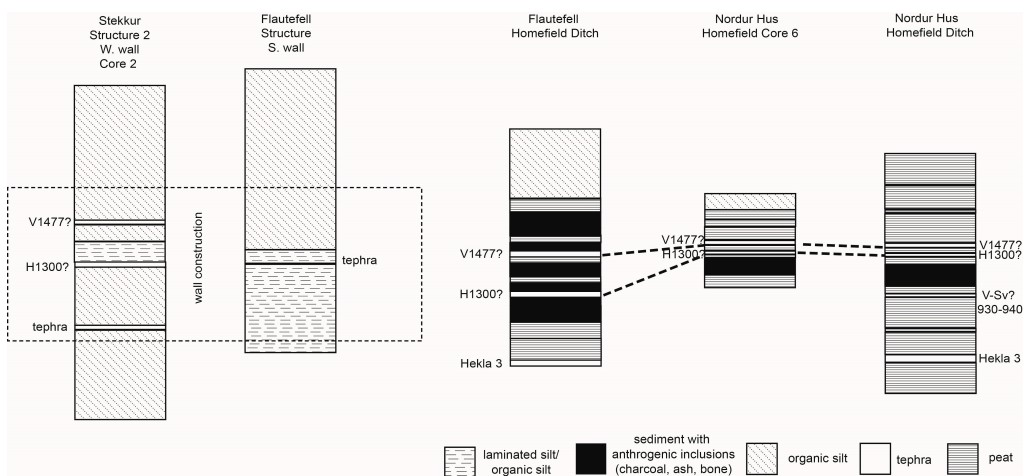

**Figure 8.** Synthesis of profiles and soils cores in abandoned turf structures and in the homefield at the NDH, FLA, and STK sites.

### 3.2.2. The Norður Hús Area (NDH)

There is very little geomorphological evidence of slope activity in the vicinity of the Norður Hús farm, which is located on a flat area and far from the mountain. No mass

movements extend into or close to the farm area; the closest is more than 500 m to the north, above the site on the Mt. Flautafell foreslope.

Furthermore, stratigraphical observations of a recently excavated drainage ditch between the farm and the mountain found no trace of coarse sedimentary transport since at least ca. 1050 B.C. (3000 B.P.), suggesting that debris flows have not affected this area in recent history. Because of its location far from the slope, only a snow avalanche of exceptional magnitude could have reached the site (it would have to cross 500 m of relatively flat ground). No traces of snow avalanches or debris-flow deposits were observed in the field. Since snow avalanches do not necessarily transport sediments, especially in their most distal part, it is possible that there are no traces to be found in the stratigraphical record. However, according to our observations, the drainage channel that discharges towards the farm could channel a very large amount of water into the farm area.

As reported in the *Jarðabók Árna Magnússonar og Páls Vídalíns*, Norður Hús was the name of a farm abandoned "long before living memory". The abandoned farm was located just to the north of the early 18th century Flautafell farm site (Table S1). According to this historical record, the farm had to be relocated because of snow avalanches and it was renamed Flutningsfell following its relocation. All of the information about this farm thus derives from local knowledge rather than from firsthand witness observations.

The *Sýslu og sóknarlýsingar Bókmenntafélagsins* suggests that the Norður Hús farm was located halfway between the river and the mountain, close to the modern farm of Flautafell. "Ruins and traces of a mountain wall" could be seen at the time the document was written (A.D. 1875). The ruins may correspond to the main ruins identified in the field, but no traces of the wall have been found. Neither the population survey nor the register of place names contains entries about Flutningsfell or Norður Hús. This helps to confirm that the farms had been occupied and abandoned well before A.D. 1712 when the first population and farm census was conducted. Modern local farmers had some knowledge of the abandoned farm but had no name for the farmstead. They also associated the ruins with the presence of a ghost, which adds some reinforcement to the hypothesis that the farm was indeed affected by a tragic event.

The stratigraphic data from the homefield of Norður Hús show some evidence of recurrent occupation of the site. The soil column comprises a series of organic matter layers of varying colors showing variable soil conditions in the area resulting from different environmental and anthropic conditions. Anthropogenic features (e.g., charcoal, wood and peat ash) were only observed in a relatively thin horizon underneath a tephra identified in the field as H1300 (Figure 8). This observation would imply an early occupation and abandonment of the farm at Norður Hús. This conclusion is consistent with the absence of the farm from population surveys and land registers, which only began during the early 18th century.

Because the Norður Hús ruin has yet only seen an initial field survey, further archeological investigation and dating of anthropogenic deposits is required to accurately document the history of occupation at this site.

### 3.2.3. The Stekkur Area (STK)

Cross-sections excavated in the distal, medial, and apical parts of the cone visible from the *Stekkur* site reveal alternating layers of very coarse massive sediments and paleosols (Figure 7). Cross-sections S04 and S07 are both located in abandoned channels on the surface of the cone. S04 is in a relic channel trending towards the structures, while S07 is close to the active part of the cone. The sections consist of alternating fine silty sand units and coarse blocky units ranging from clast-supported to matrix-supported (Figure 7). Cross-sections include a coarse blocky unit near the surface above a dark tephra layer. Tephrochronology was used to date these debris-flow deposits to after A.D. 1477 (Figure 7). This dating also coincides with the periods of activity at the nearby Flautafell site.

Section S08 is in a drainage ditch that surrounds one of the structures. It consists of 20 cm of brown and reddish-brown silty sand (Facies I). At a depth of between 35 and

10 cm, this facies integrates three dark tephra layers on top of a lens of coarse sand and subrounded gravel (Figure 7). Section S09 was excavated a few meters upslope from one of the ruins (Figure 1). It consists of fine sediments only. The absence of coarse deposits upslope from the ruins indicates that the buildings were not affected by coarse material debris flows. Thus, the slope activity at this site presents significant spatial variations that caused some areas of the cone to remain stable while others were active.

Abandoned channels are directed toward the site, indicating that sedimentary transport and a significant amount of water would have reached the site in the past. In fact, many archaeological structures are surrounded by drainage ditches that were made by the occupants in order to remove the water. The cone as a whole would have been subject to runoff events, but the archaeological structures are on the southeastern side, while the principal channel trends to the northeast. This gave some degree of protection to the structures and the fields used for hay production.

## 4. Discussion

In this study, we used geomorphological and historical methods of investigation to assess the potential impacts of slope processes on farm occupation and abandonment. Our results demonstrate that slope processes have been active in the study area and that they have had both positive and negative impacts on land use. Slope processes have been responsible for damaging fields and may have forced the abandonment of certain farms, but the slope conditions also created favorable areas for human activity. For example, the combination of the fine sediments and wet conditions provided quality grazing areas.

The slopes on Mt. Flautafell have been active since at least 1050 B.C. (3000 B.P.), and probably since deglaciation. Geomorphological and chronostratigraphical data also show evidence of slope processes close to the abandoned Flautafell farm. After reviewing the evidence, it is likely that the two channels found along the western wall are associated with the snow avalanche events described in the historical archives. The chronostratigraphical data and the census and historical records indicate that this episode occurred sometime between A.D. 1477 and 1700. No casualties were reported following this event and it does not appear to have reached an inhabited part of the Flautafell farm, although more archeological investigation is required. Perhaps the slope activity would not have directly caused the abandonment of the Flautafell farm, but it would have had a negative impact on land use.

At the *Norður Hús* site, there is little evidence to suggest that the farm was affected by snow avalanches or debris flows apart from the historical evidence. A few hypotheses could explain this: first, it is possible that the location of the farm at Norður Hús described in the archives is different from the site we studied. Without any precise indications of the location of the farmstead, its location remains uncertain. Second, the snow avalanche episode that destroyed the historical farm may have transported little to no debris, leaving no trace in the stratigraphical record.

At the *Stekkur* site, slope processes seem to have had no negative impact on land use or human occupation. In fact, distal colluvial cones could have stimulated grass growth because of the accumulation of fine sediments with good drainage, which provides suitable conditions for animal grazing. This favorable environment explains the presence of outbuildings used for animal pens and fodder storage. Even if the location made the site theoretically vulnerable to debris flows, there is no geomorphological or stratigraphical evidence to support the occurrence of such events since colonization. Due to the episodic nature of events affecting soil use in the area, it still represented high-quality summer grazing grounds for local farmers that needed little to no maintenance. Nonetheless, the active hydrological conditions on the colluvial cone may have periodically flooded the structures. The presence of a ditch around one of the structures, which is uncommon in the area, suggests that it was made to protect against waterflow and flooding. The change in hydrological conditions may have prompted modifications (entrenchment) or abandonment of the site. However, due to the lack of evidence, we cannot endorse either

hypothesis. Abandonment of these installations could also be related to external factors such as the abandonment of the site's parent farm or modifications to pastoral activity. Coring of the structures showed no evidence of occupation at the site. The soil column is sterile, formed only of massive or laminated silt originating from sheetflow and aeolian processes. The absence of tephras in turf blocks suggests it is likely to be a post-medieval site.

Given our results, we propose the following sequence of occupation: The Norður Hús farm was occupied and abandoned before A.D. 1300 because of potential snow avalanche activity or its secondary effects on field use and productivity. It was then relocated and renamed Flutningsfell or "moving mountain". Over time, the name of the farm shifted to its modern name: Flautafell. Even after its relocation, the farm was vulnerable to snow avalanche and debris flows. Between A.D. 1477–1700, the site was affected by several slope processes. In one of these events, some parts of the field were destroyed. However, it is difficult to determine if these episodes of slope activity caused the abandonment of the Flautafell farm.

The presence of farms in vulnerable areas might suggest that the population had little knowledge of slope processes and the landscape features resulting from these processes. However, the fact that the destruction of Norður Hús was recorded in archives more than 400 years after it occurred suggests that local populations had some knowledge and memory of the hazards in the area. The unpredictability of extreme weather events and the resulting slope activity may explain why farms were established in hazardous areas. Moreover, as explained by Dugmore et al. [20], the risk–benefit assessment of environmental hazards depends upon past experiences and traditional strategies of adaptation. Strategies to prevent very-low-frequency events are also prohibitive and have important social costs. It is also possible that the quality of the land was underestimated. Upland or highland grounds have often been referred to as marginal or less favorable grounds left to "underfinanced, overoptimistic fringe settlers" [22]. However, as Vésteinsson [42] demonstrated, these higher-elevation areas might have been, in fact, attractive settlement lands for the wealthy and powerful. In this case, the well-drained Flautafell talus slopes and debris cones support some of the richest grass communities in the region, which would have inevitably attracted both pastoral farmers and their flocks, especially once the available flat valley-bottom lands had reached capacity. In addition, these upland areas would have been used as summer grazing grounds, offering a transhumance option and allowing for regeneration of areas at lower lands.

Further archeological research would help to detail the chronology of occupation at the farms in our study. Although slope processes had an impact on human populations, a monocausal interpretation of these results should be avoided. Slope activity may have been an additional stressor, increasing pressure on local populations that were simultaneously experiencing the consequences of climate change (Little Ice Age), famine, epidemics, and political and socioeconomic changes.

The use of historical documents to investigate slope processes proved to be a relatively reliable source of information that complemented the geomorphological and stratigraphical data. However, the number of events recorded in the archives underestimates the actual number of events. Historically, only significant events that caused damage or death were recorded, and little to no importance was accorded to episodes that had no impact on properties. Nonetheless, our data show that even if the events were not witnessed firsthand, episodes of slope activity were preserved in local knowledge.

## 5. Conclusions

Our results validate the hypothesis that slope processes have had an impact on land use by the Norse since their establishment in the study area. Comprehensive archeological investigation remains to be carried out in order to gain a complete understanding of the history of occupation and abandonment of the farms. However, the main results of this study are as follows:

1. The field at the Flautafell farm was at least partially destroyed by slope activity between A.D. 1477 and 1700. Although there is no evidence to suggest that slope events destroyed buildings or resulted in human casualties, their occurrence during the Little Ice Age would likely have been an additional stressor on the farm.

2. The historically-known farm of Norður Hús was abandoned following a snow avalanche episode. However, the precise location of the farmstead remains uncertain. Although coring evidence shows traces of early occupation (prior to A.D. 1300) at a farm site provisionally identified as Norður Hús, only a snow avalanche of exceptional magnitude would have been able to reach this site given its distance from the slope.

3. Major slope processes, such as debris flows or snow avalanches, did not have an impact on the Stekkur site. This site was likely affected by varying hydrological conditions that flooded the outbuildings. The site is likely younger than A.D. 1477 and it would have been suitable as pasture land and for hay-making purposes, but perhaps too unstable for human occupation.

4. It is reasonable to conclude that Norður Hús was the safest location for a farm site from the point of view of slope dynamics (less frequent snow avalanche and debris flow activity). Nevertheless, it was susceptible to rare catastrophic snow avalanches and constantly received water as it drained from the mountain. Flautafell and Stekkur are somewhat similar sites, but clearly more at risk due to falling debris. Both accumulated loess prior to A.D. 1477 and then their slopes became unstable, which would have made pasturing and hay-making inside and outside the home fields less productive. A significant amount of energy would also have been needed to protect these sites from natural hazards.

**Supplementary Materials:** The following supporting information can be downloaded at: https://www.mdpi.com/article/10.3390/geosciences13020030/s1, Table S1. Synthesis of the episodes of slope activity recorded in the archives.

**Author Contributions:** J.L.: Investigation, formal analysis, writing—original draft preparation; N.B.: Conceptualization, funding acquisition, resources, supervision, writing—review and editing; J.W.: Conceptualization, funding acquisition, resources, writing—review and editing; Þ.S.: Resources, writing—review and editing. All authors have read and agreed to the published version of the manuscript.

**Funding:** This research was funded by the Natural Sciences and Engineering Research Council of Canada (NSERC), grant number RGPIN-2020-06699, the Fonds de recherche du Québec-Société et culture (FRQSC), grant number 2021-SE3-283916 and by the Social Sciences and Humanities Research Council of Canada (SSHRC), grant number 435-2012-1669.

**Data Availability Statement:** Not applicable here.

**Acknowledgments:** We wish to thank our collaborators, Guðrún-Alda Gísladóttir, Uggi Ævarsson and the Fræðasetur um Forystufé (Leader Sheep Museum/Study Center) at Svalbarð for the foundation of research supporting this study, and the residents of Svalbarðshreppur for their long-standing hospitality and support. Thanks are extended to Véronique Marengère, Dorothée Dubé, Paul Adderley, Armelle Decaulne, Maggie Joyal-Fortier, and Samuel Auger for their assistance in the field, Louise Marcoux for finalizing the figures, Lawrence Burns, and the three reviewers: Vincenzo Amato and two others anonymous reviewers for providing useful comments on the text and the Centre d'études nordiques (Université Laval), and the Institute of Earth Sciences (University of Iceland) for logistical support.

**Conflicts of Interest:** The authors declare no conflict of interest.

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
