# Peer review of "Slope Dynamics in Relation to the Occupation and Abandonment of a Mountain Farm in Þistilfjörður, Northeast Iceland"

_geosciences, doi:10.3390/geosciences13020030_

Round 1

Reviewer 1 Report

Comments are made throughout the pdf - basically this is a good paper that will be of interest to readers, but there are numerous minor inconsistencies in terminology and approach that leave the reader in need of more detail or greater clarity. I have endeavoured to provide detailed notes to help with this. All notes are intended to help make this a more consistent and readable paper, as I think the topic is of interest, but the inconsistencies and occasional lack of clarity are a little problematic. 

Author Response

Reviewer 1 Comments and suggestions

Comments are made throughout the pdf - basically this is a good paper that will be of interest to readers, but there are numerous minor inconsistencies in terminology and approach that leave the reader in need of more detail or greater clarity. I have endeavoured to provide detailed notes to help with this. All notes are intended to help make this a more consistent and readable paper, as I think the topic is of interest, but the inconsistencies and occasional lack of clarity are a little problematic. 

Response

Red text in the manuscript

As suggested by the reviewer, we removed the sentence from page 3 of 19 (According to Dupont-Hébert….).

Page 6 of 19:  higher resolution images have been added.

Page 7 of 19. The terms proximal, median and distal are commonly used in geomorphology (see for example, Cad et al. Journal of Volcanology and Geothermal Research 104 (2000) 201-235, Allen et a;, 2013 Sedimentology (2013) 60, 102–130).

In the text and in figure 4, we changed avalanche boulder tongue to scree boulder tongue.

Page 11 of 19: the legends and titles for figures 6 and 7 have been corrected.

Page 11 of 19: Landslides were mentioned (Jarðabók Árna Magnússonar og Páls Vídalíns) and they can be linked to mass movements such as avalanches.

Page 13 of 19: “The stratigraphic data from the homefield of Norður Hús show some evidence of recurrent occupation of the site. The soil column comprises a series of organic matter layers of varying colors showing variable soil conditions in the area resulting from different environmental and anthropic conditions.

Contrary to Reviewer 1’s viewpoint, we believe that this information is not speculative because we identified a succession of soil layers containing charcoal, as explained in the text.

Author Response

Response to Reviewer 2 Comments

Green text in the manuscript

Page 1 of 19: As suggested by the reviewer, we added references for the geology and geomorphology of Iceland.

Page 2 of 19. We added references indicating that slope processes are common in other areas of the world with similar climate conditions.

Page 7 of 19. As suggested by the reviewer, we added the geomorphological methods used for this study.

Pages 8 of 19 and 9 of 19. The geomorphological results presented in 3.1 are already shown in Figure 4; we improved the quality of this figure so that the results illustrations are now visible.

Reviewer 3 Report

The paper presents a very interesting case study in northern Iceland, which shows how slope processes influence land use over time. Archaeological interpretation and soil analysis (including absolute dating) are combined in this study. The paper is very well thought out and structured. The language is appropriate and the images are informative. The results are of wide interest, beyond the case study, and the conclusions help understanding the correlation between slope dynamics and human occupation in upland environments. 

Although the inferences provided by the authors are perfectly justified by the data, I would expect some reflections on the role of farming practices in accelerating or mitigating the slope processes, if there is any evidence of that. Furthermore, there seems to be an interesting temporal correlation between the intensification of these processes and the onset of the Little Ice Age. The authors mention this event, which might have affected the resilience of local communities, but they do not correlate it with the landslide and avalanche events discussed in the text.

On a further, minor, note, it might be useful to have all the dates provided as BP or BC/AD.

All that said, I think this is an excellent paper worth of publication.

Author Response

Response to Reviewer Comments and suggestions

Reviewer 3 Comment:

The paper presents a very interesting case study in northern Iceland, which shows how slope processes influence land use over time. Archaeological interpretation and soil analysis (including absolute dating) are combined in this study. The paper is very well thought out and structured. The language is appropriate and the images are informative. The results are of wide interest, beyond the case study, and the conclusions help understanding the correlation between slope dynamics and human occupation in upland environments. 

Although the inferences provided by the authors are perfectly justified by the data, I would expect some reflections on the role of farming practices in accelerating or mitigating the slope processes, if there is any evidence of that. Furthermore, there seems to be an interesting temporal correlation between the intensification of these processes and the onset of the Little Ice Age. The authors mention this event, which might have affected the resilience of local communities, but they do not correlate it with the landslide and avalanche events discussed in the text.

On a further, minor, note, it might be useful to have all the dates provided as BP or BC/AD.

All that said, I think this is an excellent paper worth of publication

Response:

We thank the reviewer for these comments. Unfortunately, at the moment, we do not have any evidence to discuss the role of farming practices in accelerating or mitigating the slope processes.

As suggested, all dates are now presented as BC/AD.

Round 2

Reviewer 2 Report

dear authors, thank you for taking all the suggestions by reviewers. the manuscript is now well-written and supported by useful and good-quality illustrations. also new references were added to the reference-list, improving the readibility of the manuscript and completing the thematic literature on geomorphology of Iceland and climatic change effects. 

thank you and good luck. Vincenzo Amato

Author Response

Dear colleague,

Thank you for providing useful comments and for your support.

Sincerely